# Simultaneous evolutionary expansion and constraint of genomic heterogeneity in multifocal lung cancer

Pengfei Ma[1], Yujie Fu[2], Mei-Chun Cai[3], Ying Yan[4], Ying Jing[1], Shengzhe Zhang[1], Minjiang Chen[5], Jie Wu[6], Ying Shen[7,8], Liang Zhu[7,8], Hong-Zhuan Chen[7,8], Wei-Qiang Gao[1], Mengzhao Wang[5], Zhenyu Gu[4], Trever G. Bivona[9,10], Xiaojing Zhao[2] & Guanglei Zhuang[1,11]

Recent genomic analyses have revealed substantial tumor heterogeneity across various cancers. However, it remains unclear whether and how genomic heterogeneity is constrained during tumor evolution. Here, we sequence a unique cohort of multiple synchronous lung cancers (MSLCs) to determine the relative diversity and uniformity of genetic drivers upon identical germline and environmental background. We find that each multicentric primary tumor harbors distinct oncogenic alterations, including novel mutations that are experimentally demonstrated to be functional and therapeutically targetable. However, functional studies show a strikingly constrained tumorigenic pathway underlying heterogeneous genetic variants. These results suggest that although the mutation-specific routes that cells take during oncogenesis are stochastic, genetic trajectories may be constrained by selection for functional convergence on key signaling pathways. Our findings highlight the robust evolutionary pressures that simultaneously shape the expansion and constraint of genomic diversity, a principle that holds important implications for understanding tumor evolution and optimizing therapeutic strategies.

[1] State Key Laboratory of Oncogenes and Related Genes, Renji-Med X Clinical Stem Cell Research Center, Ren Ji Hospital, School of Medicine, Shanghai Jiao Tong University, Shanghai 200240, China. [2] Department of Thoracic Surgery, Ren Ji Hospital, School of Medicine, Shanghai Jiao Tong University, Shanghai 200127, China. [3] State Key Laboratory of Oncogenes and Related Genes, Shanghai Cancer Institute, Ren Ji Hospital, School of Medicine, Shanghai Jiao Tong University, Shanghai 200032, China. [4] GenenDesign Co., Ltd, 590 Ruiqing Road, Shanghai 201201, China. [5] Department of Respiratory Medicine, Peking Union Medical College Hospital, Peking Union Medical College, Chinese Academy of Medical Sciences, Beijing 100730, China. [6] Department of Pathology, The Affiliated Hospital of Qingdao University, Qingdao 266000, China. [7] Department of Pharmacology and Chemical Biology, Shanghai Jiao Tong University School of Medicine, Shanghai 200025, China. [8] Shanghai Collaborative Innovation Center for Translational Medicine, Shanghai 200025, China. [9] Department of Medicine, University of California, San Francisco, California 94158, USA. [10] Helen Diller Family Comprehensive Cancer Center, University of California, San Francisco, California 94158, USA. [11] Shanghai Key Laboratory of Gynecologic Oncology, Ren Ji Hospital, School of Medicine, Shanghai Jiao Tong University, Shanghai 200127, China. Pengfei Ma, Yujie Fu and Mei-Chun Cai contributed equally to this work. Correspondence and requests for materials should be addressed to X.Z. (email: zhaoxiaojing@renji.com) or to G.Z. (email: zhuangguanglei@gmail.com)

A prevailing view of cancer holds that it is an evolutionary disease initiated by sporadic oncogenic alterations that cause pre-malignant cells to propagate faster and better survive stress than normal cells, resulting in tumor establishment[1, 2]. Many established tumors are genetically unstable, creating a tendency to accumulate additional mutations; as a result, the constituent malignant cells within a tumor are constantly evolving[3, 4]. One consequence of these forces is substantial genomic divergence that causes extensive intratumoral heterogeneity and branching phylogenies[5–10]. Cumulative evidence is emerging that the genetic complexity and molecular evolution of a tumor profoundly impact the clinical course of individuals who suffer from many different cancer types[11, 12].

One of the pivotal, yet largely under-investigated aspects in the framework of this general scheme is to what extent the genomic diversity present in a tumor contributes to phenotypic heterogeneity[13]. It has been generally accepted that the myriad of cancer genetic repertoires can permit phenotypic plasticity, allowing the tumor to dynamically adapt to local and systemic pressures including those exerted by treatments[11, 14, 15]. However, it is still possible that under a given circumstance evolutionary constraints may apply to restrict tumor genetic drivers to limited options and lead to phenotypic convergence. There has been preliminary evidence to support this hypothesis. For example, different tumor subclones or distinct metastatic sites were occasionally shown to undergo parallel evolution of the same gene, pathway or protein complex during carcinogenesis or drug treatment[16–19]. Similarly, four independent tumors occurring in a patient with Von Hippel Lindau syndrome exhibited functionally recurrent activation of the mTOR pathway[20]. One caveat of these studies is that the tumor clones or subclones studied shared largely identical somatic alterations or well-known germline cancer-predisposition genes, which may potentially define an intrinsic boundary for evolutionary divergence. From a therapeutic perspective, an improved understanding of the competing forces controlling on the one hand the expansion, and on the other hand the constraint of genomic diversity and heterogeneity during tumor evolution is essential to precisely decipher the Achilles' heels of human cancers for rational intervention.

MSLC is diagnosed as multiple tumor nodules in the same or different lung lobes[21, 22]; thus, MSLCs by definition share an identical germline genetic background and environmental exposure in individual patients. MSLC is a frequent occurrence in individuals with lung cancer[23–25], yet the molecular origins and relationships among the synchronous tumors remain largely unclear in most patients[26–28]. We reasoned that the genomic analysis of MSLC to determine the extent of common versus divergent ancestry, an under-investigated area to date, would offer a unique opportunity to provide insights into the evolutionary principles that shape tumorigenesis.

In this study, we present a detailed genetic and experimental analysis on a collection of clinical specimens from patients with MSLCs. Whole-exome sequencing reveals independent clonality of synchronous lesions in each individual and profound genomic heterogeneity at both interfocal and intrafocal levels. However,

the functional convergence of distinct oncogenic drivers on key signaling pathways within multicentric primary tumors is evident as a possibly generalizable principle that can aid rational therapeutic selection. Therefore, simultaneous evolutionary expansion and constraint of genomic heterogeneity may competitively and collaboratively shape tumorigenesis of lung cancer and potentially other human malignancies.

## Results

**Clonal architecture and mutational landscape of MSLC.** To gain further insights into the genetic alterations shaping lung tumorigenesis, we sampled four patients (RJLC1-4; Table 1) with treatment-naive MSLC (16 tumor samples within 11 independent lesions, all adenocarcinomas) and performed whole-exome sequencing on tumor and matched adjacent normal lung DNA (Supplementary Fig. 1), yielding a mean coverage of 244× (205×−296×). Approximately 99% of targeted bases were covered to a depth of 20× or more (Supplementary Table 1). We identified 167–679 somatic alterations per tumor (Supplementary Data 1), including a total of 373 non-synonymous exonic mutations present in at least one tumor region. In order to determine whether MSLC was derived from one single lesion with intra-thoracic metastases or multiple local primary tumors, non-silent mutations were mapped on the basis of their intertumoral distributions and classified as shared (present in more than one tumor) or private (present in only one tumor) alterations. In addition, we constructed phylogenetic trees to estimate the ancestral relationships of individual foci (Fig. 1a). In agreement with a recent study demonstrating independent clonal origins of MSLCs[28], significantly overlapping variant sets were absent between any tumor pair in each of these four patients. As a comparison, multi-region sequencing of two larger lesions (RJLC1-T1 and RJLC4-T1) revealed many shared somatic mutations among sequenced samples within the same tumor (Fig. 1a). This observation suggests that multiple pulmonary nodules within one subject probably arose autonomously from different progenitor cells.

The mutation spectra of single-nucleotide variations (SNVs) were fairly consistent across lung cancers from the same individuals; however, notable discordance was observed among the three tumors of patient RJLC3. The majority of sequenced samples displayed a preponderance of C > T and T > G transversions, but not C > A transitions associated with tobacco exposure[29], consistent with the non-smoking history of these patients (Fig. 1b). We detected genetic aberrations involving genes previously reported to be recurrently altered in lung adenocarcinoma[30], including EGFR, KRAS, MET, BRAF, and TP53. Other identified putative driver mutations often affected genes regulating transcription, MAPK signaling, cell adhesion and survival (Fig. 1b; Supplementary Table 2). Overall, few driver mutations were shared between any two tumors, further suggesting that our MSLC cohort represented independently arising primary tumors. Detailed analysis of mutational signatures[31] separated all specimens into two major clusters and indicated that tumors from the same patient usually clustered

---

**Table 1 Patient characteristics in the MSLC cohort**

| Patient ID | Age | Gender | Ethnicity | Smoking history | Histology | TNM stage | Tumor sites (n) |
|---|---|---|---|---|---|---|---|
| RJLC1 | 64 | Female | Han Chinese | Non-smoker | AIS | pTisN0M0 | RUL(1) RLL(1) |
| RJLC2 | 52 | Female | Han Chinese | Non-smoker | AIS | pTisN0M0 | RUL(3) |
| RJLC3 | 57 | Female | Han Chinese | Non-smoker | AIS | pTisN0M0 | LUL(3) |
| RJLC4 | 60 | Female | Han Chinese | Non-smoker | ADC | pT1N0M0 | RUL(1) LUL(1) LLL(1) |

ADC adenocarcinoma, AIS adenocarcinoma in situ, LLL left lower lobe, LUL left upper lobe, RLL right lower lobe, RUL right upper lobe

---

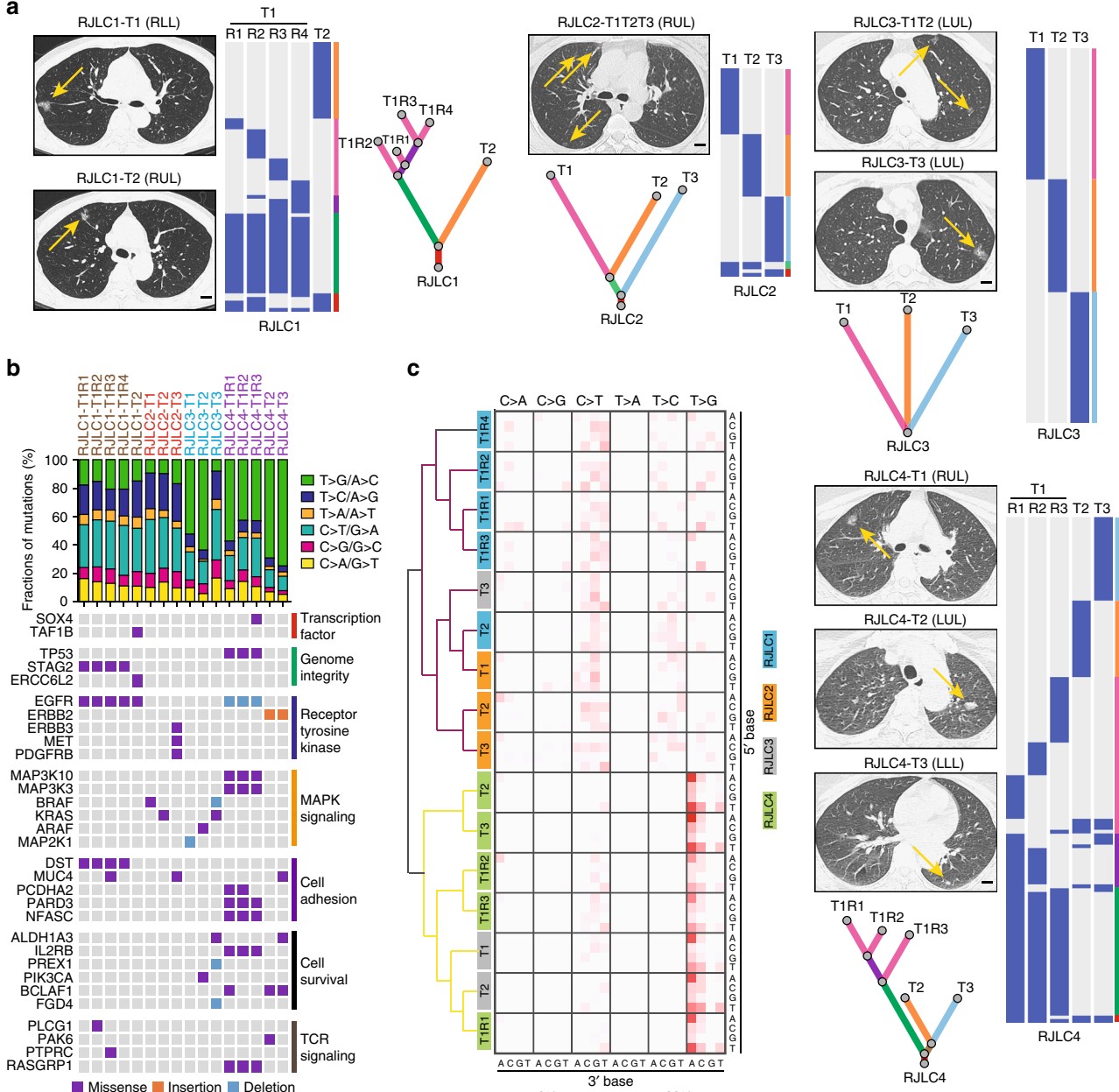

**Fig. 1** Clonal architecture and mutational landscape of MSLC. **a** Computed tomography (CT) diagnosis and clonal architecture of multifocal tumors in four MSLC patients. Heat maps showed the presence (blue) or absence (gray) of all non-silent somatic mutations in every tumor region. Phylogenetic trees indicated the clonal structure of sequenced tumor regions in each patient. Scale bar, 1 cm. **b** Mutational landscape of all 16 sequenced tumor regions. Putative driver genes with somatic mutations were classified according to the functional categories. **c** Frequencies for each of the six substitutions at all 16 possible trinucleotide contexts were displayed in a heat map. All specimens were separated into two major clusters based on the mutational signatures

together with the exception of RJLC3 (Fig. 1c; Supplementary Fig. 2). This finding suggests that distinct mutational processes were operating to promote tumorigenesis within different lesions in this case. We further used the deconstructSigs framework to extract known mutational signatures that might contribute to the specific mutational profiles in each cluster[32, 33]. Interestingly, cluster 1 exhibited an enrichment of signature 1 (age-related), signature 3 (associated with failure of DNA double-strand break-repair by homologous recombination), signature 9 (attributable to polymerase η and implicated with AID activity), and signature 12 (unknown etiology). In striking contrast, cluster 2 was almost exclusively characterized by signature 9 (Supplementary Fig. 3), which had been described only in chronic lymphocytic leukemia

and malignant B-cell lymphoma and not previously in lung cancer. Additional investigation is required to determine the mechanistic underpinnings and biological significance of these findings.

**Interfocal and intrafocal genetic heterogeneity of MSLC.** In-depth genomic sequencing of MSLC enabled multi-layer delineation of tumor heterogeneity at interpatient, interfocal and intrafocal levels (Fig. 2a). In accordance with the genetically heterogeneous TCGA lung cancers[30], each MSLC patient had a unique set of somatic variants, as exemplified by a distinct mutation spectrum and largely non-overlapping exonic

substitutions (Fig. 2b). Analysis of genetic (germline) predis-position also failed to uncover meaningful single-nucleotide polymorphisms (SNPs) associated with lung cancer susceptibility, further highlighting the likely role of distinct somatic genetic alterations in driving the genesis of each tumor.

To further corroborate the interfocal diversification revealed by phylogenetic analysis, we functionally inferred the most promi-nent cancer-driving abnormality in each tumor (Supplementary Table 3; Supplementary Figs. 4–7). *KRAS*, a well-known proto-oncogene, was mutated in RJLC2-T2 (KRAS$^{G12D}$) and RJLC3-T3

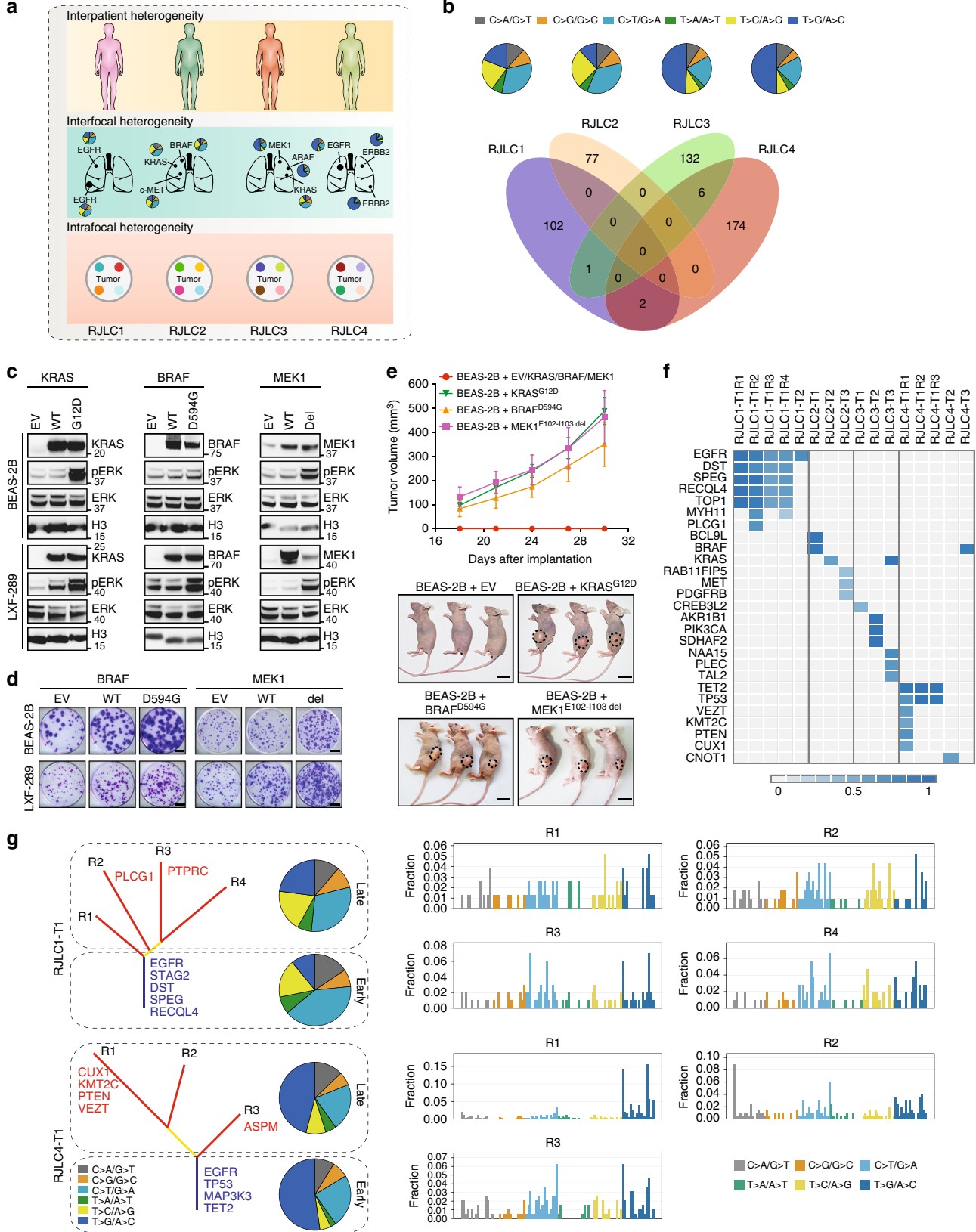

(KRAS$^{G12C}$), which presumably resulted in constitutive MAPK signaling. Similarly, RJLC1-T1, RJLC1-T2, and RJLC4-T1/T2/T3 harbored activating *EGFR* mutations (EGFR$^{L858R}$ or EGFR$^{E746-A750\ del}$) or an *ERBB2* in-frame insertion (HER2$^{YMVA}$) to potentiate oncogenic transformation. A *MET* mutation causing exon 14 skipping in RJLC2-T3 (c-MET$^{\Delta E14}$), and ARAF$^{S214C}$, identified in RJLC3-T2, have been recently reported to be oncogenic in preclinical models and lung adenocarcinoma patients[34–36]. Indeed, we were able to experimentally validate the tumorigenic potential of ARAF$^{S214C}$, c-MET$^{\Delta E14}$, and HER2$^{YMVA}$ (Supplementary Fig. 8a). Notably, alterations were detected in both *ERBB3* (also known as HER3) and *PDGFRB* genes in RJLC2-T3 but we found that the variants did not lead to substantial mutant-specific AKT or ERK hyperactivation (Supplementary Fig. 8b). However, we noted two candidate clinically relevant alterations located in protein kinase domains (Supplementary Fig. 8c), BRAF$^{D594G}$ in RJLC2-T1 and MEK1$^{E102-I103\ del}$ (encoded by *MAP2K1*) in RJLC3-T1. Each mutant, despite being previously discovered in various cancer types, had not been explicitly implicated in driving lung malignancy. To characterize the functional impact of the BRAF and MEK1 mutations, we ectopically expressed mutant alleles in a human embryonic kidney cell line HEK293T, an immortalized bronchial epithelial cell line BEAS-2B and a lung adenocarcinoma cell line LXF-289, using the KRAS$^{G12D}$ oncogenic variant as a positive control. Interestingly, BRAF$^{D594G}$ exhibited attenuated catalytic activity toward downstream MAPK pathway in HEK293T (Supplementary Fig. 8d), consistent with its location in the highly conserved DFG motif and previously reported kinase-inactivating nature[37]. However, the transforming capability of BRAF$^{D594G}$ was evident in BEAS-2B or LXF-289 cells presumably in a RAS/CRAF-dependent manner[37], as assessed by phospho-ERK, colony-forming and xenograft tumorigenicity assays (Fig. 2c–e). On the other hand, MEK1$^{E102-I103\ del}$ clearly acted as a biochemical gain-of-function mutation in all three cell lines (Fig. 2c–e) and was associated with response to the MEK inhibitor trametinib in vivo (Supplementary Fig. 8e). Importantly, we established a patient-derived lung tumor xenograft model harboring BRAF$^{D594N}$, which likewise enhanced colony formation relative to vector control or wild-type BRAF (Supplementary Fig. 8f). We observed a remarkable sensitivity to MAPK pathway targeted therapies, particularly the combination of dabrafenib and trametinib (Supplementary Fig. 8g). These data collectively reveal the oncogenicity and targetability of BRAF$^{D594G}$ and MEK1$^{E102-I103\ del}$ in lung cancer. Therefore, each multicentric primary tumor could be driven by distinct molecular machinery and interfocal heterogeneity was highly prevalent in MSLC. Furthermore, although all MSLCs exhibited positive staining of CD3 (Supplementary Fig. 9a), indicating that T cells already infiltrated into tumors even in early-stage lung cancer, each lesion was predicted to harbor distinct repertoires and unequal load of neoantigens (Supplementary Fig. 9b; Supplementary Table 4), suggestive of functionally heterogeneous tumor-associated lymphocytes and potentially differential clinical responses to immune treatment regimens.

To estimate the extent of intrafocal heterogeneity in MSLC, we first performed single-sample clonality analysis using PyClone[38] (Supplementary Fig. 10) and SciClone[39] (Supplementary Fig. 11). Both methods consistently indicated that the majority of sequenced specimens were oligoclonal (Supplementary Table 5). Nevertheless, the key putative driver mutations were proven to be mostly clonal on the basis of cancer cell fraction (the fraction of tumor cells harboring the SNVs) (Fig. 2f), consistent with previous reports showing that driver mutations occur early during lung cancer development[8]. Additionally, two larger tumors (RJLC1-T1 and RJLC4-T1) were subjected to multi-region sequencing (M-seq), permitting a comprehensive dissection of spatiotemporal intrafocal evolution. Corroborating prior observations[7, 8], phylogenetic reconstruction of the M-seq data yielded evidence of a branched evolutionary pattern (Fig. 2g). The ITB (indices of branched to truncal mutations) for these two tumors were generally higher compared with those of lung adenocarcinomas analyzed by M-seq recently (Supplementary Table 6). The spectra of point mutations in each tumor displayed considerable temporal difference between early (truncal) and late (branched) SNVs. Consistent with previous reports[7, 8], a relatively higher percentage of somatic mutations could be ascribed to APOBEC-mediated mutagenesis on the branches compared with the trunk in both tumors (Supplementary Fig. 12). Moreover, spatial heterogeneity, as indicated by divergent mutational signatures, was evident among geographically separated tumor regions (Fig. 2g). Together, these findings suggested that dynamic mutational processes shaped subclonal genomes over time, conceivably contributing to the substantial intrafocal heterogeneity observed in MSLC.

**Constrained tumorigenic pathways among multicentric lesions.** We noted a striking biological convergence of heterogeneous driver events among individual foci on the same signaling pathway in each MSLC case. For example, RJLC1 and RJLC4 both altered the EGFR family of receptor tyrosine kinase, whereas RJLC2 and RJLC3 tended to primarily impact on the MAPK pathway (Fig. 3a; Supplementary Fig. 13a). While this phenomenon was reminiscent of a previously described parallel evolution principle[11, 40], the functional forces that underlie this striking biological convergence and its consequences on tumorigenesis and treatment response remain largely unresolved. For instance, an important unanswered question in the field is whether individual, distinct genetic alterations that are present in different tumors impart identical or disparate signaling outputs or treatment responses. We therefore used this unique MSLC system to test the hypothesis that different driver mutations in each patient are functionally interchangeable, a principle that has been implied but not formally demonstrated to date in lung cancer (to our knowledge). We designed an experimental workflow to specifically control the lung cancer cell population using the clustered regularly interspaced short palindromic repeats (CRISPR)-Cas9 system (Fig. 3b). To this end, EGFR$^{E746-A750\ del}$ was knocked out in PC9 cells, which resulted in decreased phospho-AKT and phospho-ERK. The EGFR-null PC9 cells could only

**Fig. 2** Interfocal and intrafocal genetic heterogeneity of MSLC. **a** A schematic overview of tumor heterogeneity analysis at interpatient, interfocal and intrafocal levels. **b** Mutation spectra of four MSLC patients were summarized by pie charts and shared mutations between different subjects were showed in Venn diagram. **c** Overexpression of indicated wild-type or mutant genes in BEAS-2B or LXF-289 cells. Gene expression and ERK phosphorylation were measured by immunoblotting. **d** Indicated wild-type or mutant genes were lentivirally introduced into BEAS-2B or LXF-289 cells. Cells were cultured for two weeks and stained with crystal violet. Scale bar, 5 mm. **e** Tumor growth of BEAS-2B xenografts that ectopically expressed indicated mutant genes. Each line represented mean tumor volume of the respective group, and error bars indicated standard deviation (10 mice per group). Scale bar, 10 mm. **f** A heat map presented the CCF of putative driver mutations in each sequenced region of the MSLC tumors. **g** Mutation spectra of early mutations (trunk) and late mutations (branch), and mutational signatures of different tumor regions in M-seq samples of RJLC1-T1 and RJLC4-T1

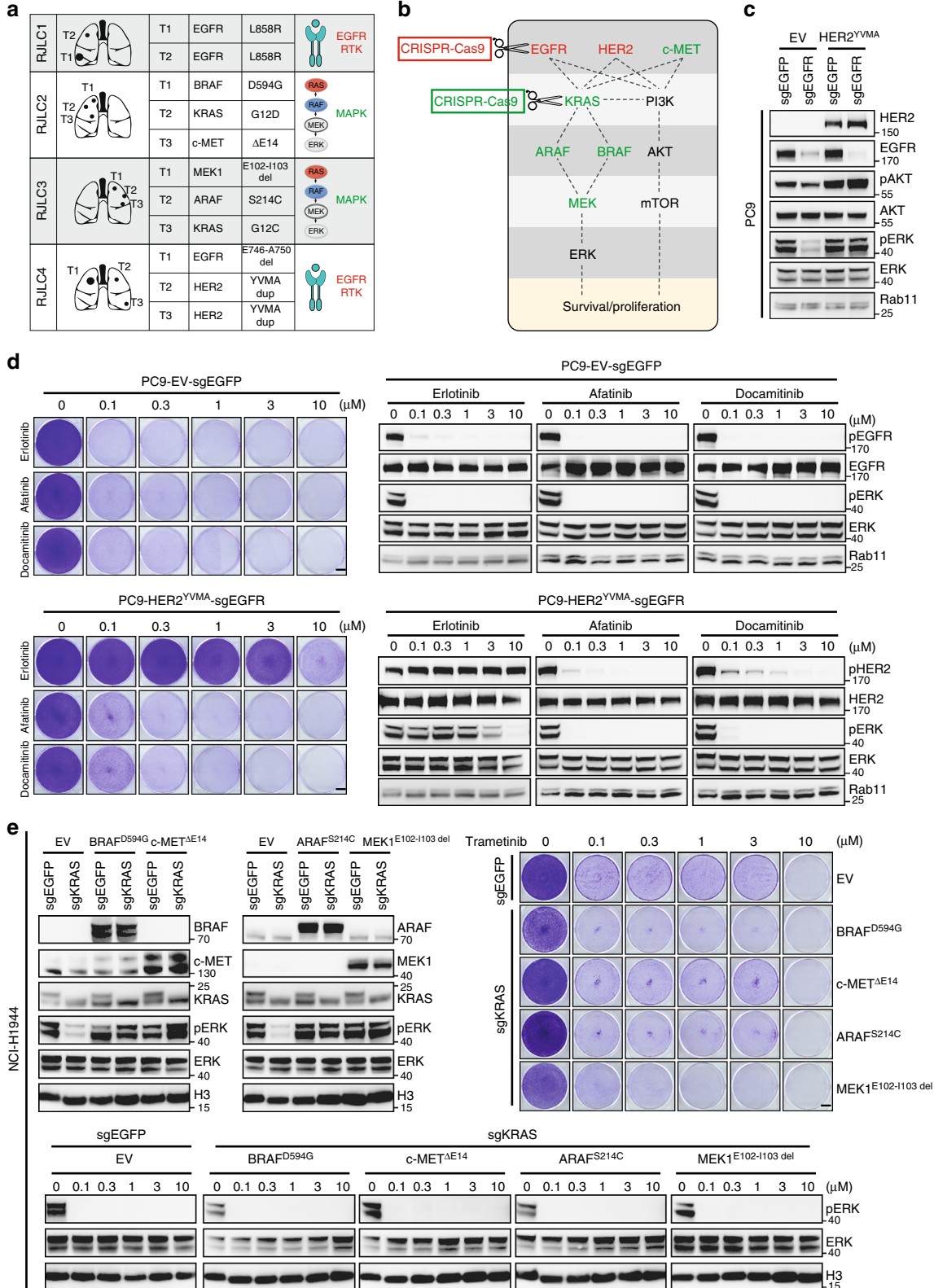

**Fig. 3** Constrained tumorigenic pathways among multicentric lesions. **a** A summary of the most prominent driver mutation identified in each primary tumor of MSLC. **b** EGFR or KRAS was knocked out using CRISPR-Cas9 system in lung cancer cell lines and replaced with HER2, c-MET, ARAF, BRAF, or MEK mutants. **c** EGFR was knocked out using CRISPR-Cas9 system in PC9 cells and replaced with HER2^YMVA. AKT and ERK phosphorylation were measured by western blot analysis. **d** Cells were treated with a serial dilution of indicated inhibitors for a week and stained with crystal violet. Scale bar, 5 mm. The corresponding cell lysates were analyzed by immunoblotting. **e** KRAS was knocked out using CRISPR-Cas9 system in NCI-H1944 cells and replaced with BRAF^D594G, c-MET^ΔE14, ARAF^S214C, or MEK1^E102-I103 del. ERK phosphorylation was measured by western blot analysis. Indicated cells were treated with a serial dilution of trametinib for a week and stained with crystal violet. Scale bar, 5 mm. The corresponding cell lysates were analyzed by immunoblotting

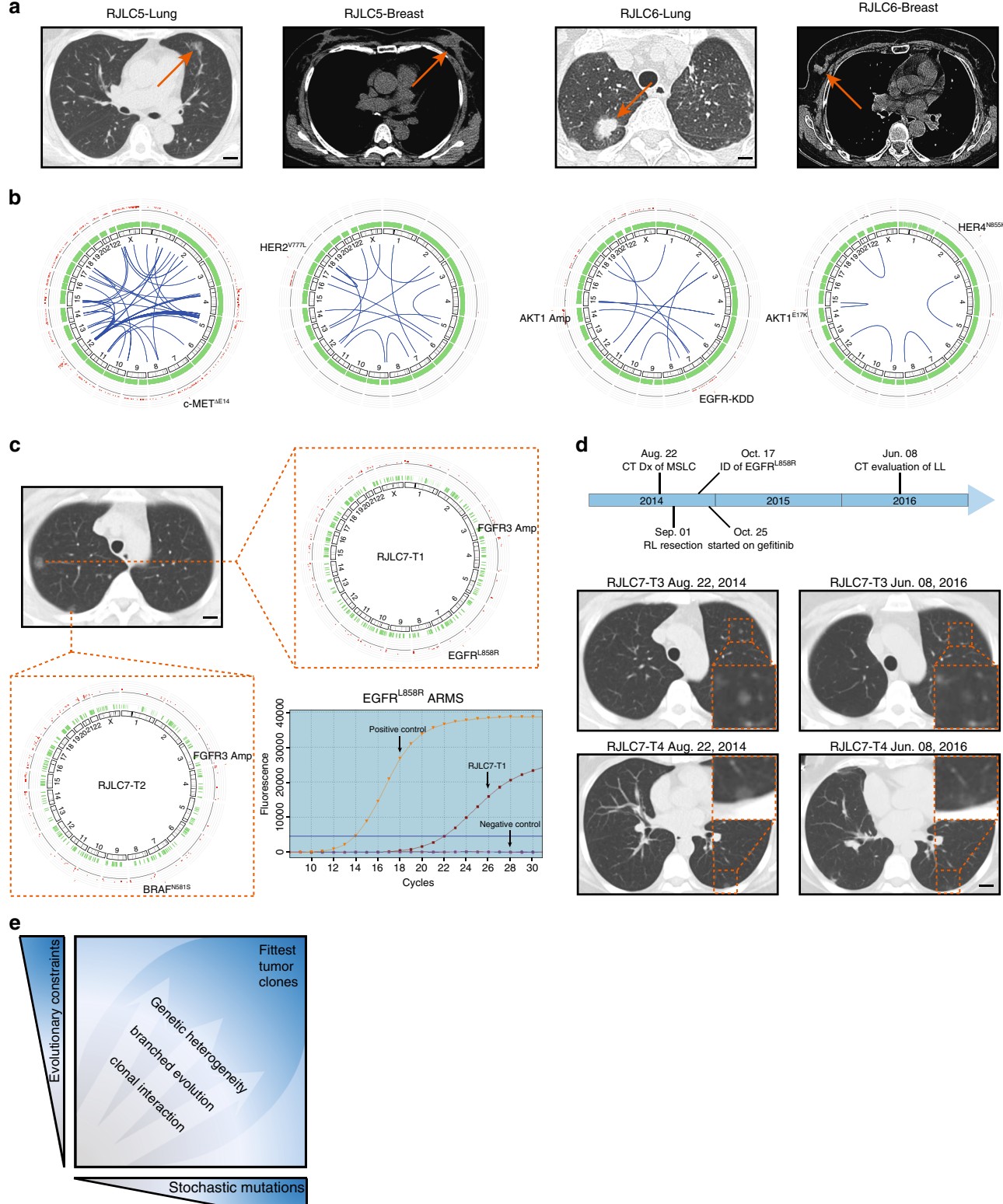

**Fig. 4** Independent validation of oncogenic pathway convergence. **a** CT diagnosis of synchronous lung and breast tumors in RJLC5 and RJLC6. Scale bar, 1 cm. **b** Circos plots highlighting different structural variations of lung and breast tumors in RJLC5 and RJLC6. RJLC5-Lung harbored c-MET$^{\Delta E14}$ and RJLC5-Breast harbored HER2$^{Y777L}$; RJLC6-Lung harbored EGFR-KDD and RJLC6-Breast harbored HER4$^{N855K}$. **c** CT diagnosis of MSLC and circos plots representation of somatic mutations in RJLC7. EGFR$^{L858R}$ identified in RJLC7-T1 was confirmed using ARMS PCR. Scale bar, 1 cm. **d** Time line of RJLC7 MSLC diagnosis and treatment. Baseline and 21-month CT scan for RJLC7 following treatment with gefitinib. Scale bar, 1 cm. **e** A schematic model showing that simultaneous evolutionary expansion and constraint of genomic heterogeneity collaboratively shape tumorigenic process and clonal architecture

survive a short period of time. When we introduced HER2$^{YMVA}$ into these cells, the impaired AKT and ERK phosphorylation and growth capacity were fully restored (Fig. 3c). Accordingly, substitution of EGFR$^{E746-A750\ del}$ with HER2$^{YMVA}$ conferred insensitivity to erlotinib, an EGFR inhibitor with less activity against HER2, but not afatinib or dacomitinib, which each inhibited HER2 kinase activity (Fig. 3d). To model RJLC2 and RJLC3, we applied this method to KRAS-mutant NCI-H1944 cells and found that BRAF$^{D594G}$, c-MET$^{\Delta E14}$, ARAF$^{S214C}$, or MEK1$^{E102-I103\ del}$ expression could indeed functionally rescue the depletion of KRAS$^{G13D}$ (Fig. 3e). Similar results were obtained with NCI-H2009 cells harboring KRAS$^{G12A}$ (Supplementary Fig. 13b). We conclude that the evolutionary trajectories of independent tumors in MSLC might converge to cause preferential activation of constrained, critical oncogenic pathways.

**Independent validation of oncogenic pathway convergence.** Several complementary lines of evidence further validated this principle of biologically constrained expansion of somatic genetic alterations in MSLC development that we uncovered. First, recent genomic sequencing of small MSLC cohorts revealed frequent mutational co-occurrence in the same cancer gene among different tumors (Supplementary Table 7), even though these tumors originated from distinct progenitor cells[8, 28]. Second, we retrospectively identified two patients (RJLC5 and RJLC6) with synchronous lung and breast cancers (Fig. 4a), who underwent surgical resection of both tumors. We collected fresh tissues for these patients and conducted whole-genome sequencing (WGS), revealing independent clonality of lung and breast tumors based on somatic mutations (Supplementary Fig. 14a). A circos plot analysis showed no common structural alterations and confirmed that these synchronous tumors developed through an accumulation of completely different genetic events (Fig. 4b). However, the two tumors in each patient, albeit from different tissues of origin, showed remarkably similar mutation spectra and signatures (Supplementary Fig. 14b). In addition, we identified c-MET$^{\Delta E14}$, HER2$^{Y777L}$, EGFR-KDD, and HER4$^{N855K}$ as the most predominant driver mutants in these four tumors (Fig. 4b; Supplementary Fig. 14c), indicative of the convergent central role of receptor tyrosine kinases in both lung and breast cancer. Finally, we were able to test the potential therapeutic implications of our findings in a pilot prospective study. Patient RJLC7 was hospitalized due to persistent acroanesthesia and was subsequently diagnosed with MSLC, presenting as a number of ground glass opacities (GGO) spread among three lung lobes. In August 2014, the patient received a video-assisted thoracoscopic surgery (VATS)-guided right upper lobectomy, right lower lobe wedge resection and lymphadenectomy. Two relatively larger lesions were removed and pathologically graded as pT1N0M0 stage IA lung adenocarcinomas. WES indicated that RJLC7-T1 and RJLC7-T2 were multicentric primary tumors (Supplementary Fig. 15a), but shared identical mutation spectra and signatures (Supplementary Fig. 15b). Both tumors harbored FGFR3 focal amplification. Additionally, we detected BRAF$^{N581S}$ in RJLC7-T2 and EGFR$^{L858R}$ in RJLC7-T1 (Fig. 4c), which was confirmed using amplification refractory mutation system (ARMS). Although there were several small nodules present as localized disease in her left upper lobe, the patient declined a secondary surgical procedure, as well as a biopsy for molecular analysis, and instead elected to begin gefitinib treatment off-label to address the possible progression of her early-stage disease. She was monitored closely with serial imaging and gradual GGO regression was observed (Fig. 4d). Although these data must be interpreted with caution, the response to the EGFR-targeted drug suggests an EGFR signaling dependency in this distinct area of multifocal

disease that is consistent with the activating EGFR$^{L858R}$ mutation found in RJLC7-T1; these findings provide additional suggestive evidence of interfocal convergent evolution in MSLC. The patient remains on therapy without notable adverse effects to date. An expanded, multicenter clinical investigation has been planned to determine the extent and significance of convergent tumor evolution in patients with MSLC.

**Discussion**

When viewed through the lens of evolutionary biology, cancer has been considered as an incessantly evolving ecosystem involving dynamic interplay between tumor cells and their microenvironment[2]. The fittest clones survive and expand, following the Darwinian rule of natural selection[41]. Moreover, genetic instability is an integral component of human neoplasia, and as a result, stochastic mutagenesis constantly takes place and may phenotypically alter clonal fitness[3]. These repeated rounds of new mutation emergence and pervasive positive and negative selection make intertumor and intratumor heterogeneity almost inevitable[42, 43]. Indeed, our study has provided genomic portraits of the interfocal and intrafocal molecular landscape of MSLC, and revealed a high prevalence of heterogeneity despite a shared genetic background and environmental exposure in each subject. These data validate and extend previous findings of pronounced intratumor heterogeneity observed in non-small cell lung cancer, which has been speculated to complicate the molecular diagnosis and therapeutic interference of this deadly disease[7, 8].

However, an important question that remains to be addressed is how much of the genetic heterogeneity will prove to be functionally relevant and clinically relevant. Interestingly, our study revealed that most key driver mutations were clonal and the seemingly distinct multifocal tumors often converged on selected oncogenic pathways that our evidence suggests are functionally interchangeable. Notably, we developed a specific experimental workflow based on the CRISPR-Cas9 system, and formally demonstrated that distinct genetic alterations harbored by different foci in each MSLC case imparted equivalent signaling outputs and treatment responses. These data therefore argue strongly for evolutionary constraints during lung cancer initiation and progression. Taken together, our observations highlight the coexistence of genomic diversity and phenotypic convergence as a possibly generalizable principle for understanding tumor evolution, which may hold potential implications for identifying and prioritizing therapeutic strategies. On one hand, the degree of genetic diversity present in many tumors and patients necessitates the functional inference of aberrant oncoprotein activity driving individual tumors or subclones, in order to apply precise pharmacological inhibition. On the other hand, the striking functional convergence of heterogeneous cancer-driving abnormalities may enable the reproducible prediction of their biological consequence and therapeutic "actionability". Importantly, we showed in a pilot study that how this concept could aid rational therapeutic prioritization against MSLC when a molecular diagnosis was not feasible. Conceivably, similar competing evolutionary forces may likewise operate during anti-cancer treatment, which, if ultimately validated in prospective clinical investigations, will help formulate strategic guidelines for the management of anticipated drug resistance.

In conclusion, our findings not only provide the rationale for employing more sophisticated genomic profiling to distinguish between potentially curable multiple primary tumors and unresectable metastatic dissemination in patients with MSLC, a disease with increasing incidence, but also provide initial evidence for perhaps an underestimated extent of parallel evolution and recurrent patterns in the spatiotemporal acquisition of driver mutations during tumorigenesis. A deeper understanding of the

dueling evolutionary forces that control the simultaneous expansion and constraint of tumor genomic diversity and heterogeneity (please refer to the schematic model shown in Fig. 4e; also see ref. [44]) promises to improve the diagnosis and treatment of lung cancer and other malignancies.

## Methods

**Patient cohort description.** The study was approved by the Ren Ji Hospital Ethics Committee. All patients provided written informed consent. Tumor samples for sequencing were obtained from five patients diagnosed with pathologically confirmed multiple synchronous lung cancers (MSLCs). The other two patients were diagnosed with synchronous lung and breast adenocarcinomas. These patients underwent surgical resection prior to receiving any form of adjuvant therapy. All samples were collected in Ren Ji Hospital, Shanghai Jiao Tong University School of Medicine. Tumor sizes ranged from 0.5 to 3.6 cm according to pathology reports. All patients were free of extrathoracic metastasis. Detailed clinical characteristics were provided in Table 1. Eighteen samples from five patients with MSLCs were subjected to whole-exome sequencing (WES), and four samples from two patients with lung and breast cancers were subjected to whole-genome sequencing (WGS).

**Tumor processing.** Up to ten 10 μm fresh frozen sections for each tumor sample and adjacent normal tissue were selected by a pathologist, documented by photography, and snap-frozen. Haematoxylin-eosin-stained slides were reviewed by experienced lung cancer pathologists to determine the histomorphological subtype and the proportion of malignant cells relative to nonmalignant stromal cells. Approximately $5 \times 5 \times 5$ mm tumor tissues were used for genomic DNA extraction, using the DNeasy kit (Qiagen) according to manufacturer's protocol. DNA was quantified by Qubit (Life Technologies) and DNA integrity was examined by agarose gel electrophoresis.

**Whole-exome sequencing.** Paired-end DNA library were prepared according to manufacturer's instructions (Agilent). Genomic DNAs (gDNA) from patients RJLC1 to 4 and RJLC7 were sheared into 180 ~ 280 bp fragments by Covaris S220 sonicator. Ends of the gDNA fragments were repaired; 3′ ends were adenylated. Both ends of the gDNA fragments were ligated at the 3′ ends with paired-end adaptors (Illumina) with a single 'T' base overhang and purified using AMPure SPRI beads from Agencourt. The adaptor-modified gDNA fragments were enriched by six cycles of PCR using SureSelect Primer and SureSelect ILM Indexing Pre Capture PCR Reverse Primer. The concentration and the size distribution of the libraries were determined on an Agilent Bioanalyzer DNA 1000 chip. Whole-exome capture was carried out using Agilent's SureSelect Human All Exon V5 Kit. An amount of 0.5 μg prepared gDNA library was hybridized with capture library of biotinylated RNA baits for 5 min at 95 °C, 24 h at 65 °C. The captured DNA–RNA hybrids were recovered using Dynabeads MyOne Streptavidin T1 from Dynal. DNA was eluted from the beads and desalted using Qiagen MinElute PCR purification columns. The purified capture products were then amplified using the SureSelect ILM Indexing Post Capture Forward PCR Primer and PCR Primer Index 1 through Index 16 (Agilent) for 12 cycles. 50 Mb of DNA sequences of 334,378 exons from 20,965 genes were captured. After DNA quality assessment, captured DNA library were sequenced on Illumina Hiseq 4000 sequencing platform (Illumina) according to manufacture's instructions for paired-end 150 bp reads (Novogene, Beijing). Libraries were loaded onto paired-end flowcells at concentrations of 14–15 pM to generate cluster densities of 800,000–900,000 per mm² using Illumina cBot and HiSeq paired-end cluster kit version 1. The sequencing depth was 200×.

**Whole-genome sequencing.** A total amount of 1 μg genomic DNA per sample was used as input material for the DNA library preparations. Sequencing library was generated using Truseq Nano DNA HT Sample Prep Kit (Illumina) following manufacturer's recommendations and index codes were added to each sample. Briefly, genomic DNA sample was fragmented by sonication to a size of 350 bp. Then DNA fragments were end-polished, A-tailed, and ligated with the full-length adapter for Illumina sequencing, followed by further PCR amplification. After PCR products were purified with AMPure XP system, libraries were analyzed for size distribution by Agilent 2100 Bioanalyzer and quantified by real-time PCR (3 nM). The clustering of the index-coded samples was performed on a cBot Cluster Generation System using Hiseq PE Cluster Kit (Illumina) according to the manufacturer's instructions. After cluster generation, the DNA libraries were sequenced on Illumina Hiseq 4000 sequencing platform (Illumina) and 150 bp paired-end reads were generated. The sequencing depth was 60×.

**Sequence alignment and variant calling.** Paired-end clean reads in FastQ format generated by the Illumina pipeline were aligned to the reference human genome (UCSC hg19) by Burrows-Wheeler Aligner (BWA) to get the original mapping results stored in BAM format[45]. SAMtools[46], Picard (http://broadinstitute.github. io/picard/), and GATK[47] were used to sort BAM files and do duplicate marking, local realignment, and base quality recalibration to generate final BAM files for

computation of the sequence coverage and depth. The somatic SNVs were detected by VarScan version 2.2.5 and MuTect[48, 49], and the somatic InDels detected by GATK Somatic Indel Detector. ANNOVAR was performed to do annotation for VCF (Variant Call Format), including querying the databases of dbSNP and the 1000 Genomes Project[50]. Variant position, variant type, conservative prediction and other information were obtained at this step. Gene transcript annotation databases, such as Consensus CDS, RefSeq, Ensembl and UCSC, were applied for annotation to determine amino acid alternations. Variants obtained from previous steps were compared against SNPs present in the dbSNP and 1000 Genomes databases (1000 Genomes Project Consortium) to discard known SNPs. Only SNVs occurring in exons or in canonical splice sites were further analyzed. The retained non-synonymous SNVs were submitted to PolyPhen and SIFT for functional prediction. Control-FREEC was utilized to detect somatic CNV[51]. For samples from patient RJLC5 and RJLC6, which were subjected to WGS, breakdancer was implemented to identify potential structural variants[52]. All these variant results were visualized using Circos[53].

**Identification of putative driver mutations.** All identified non-silent mutations were compared with lists of potential driver genes in NSCLC, containing all genes identified as frequently mutated by large-scale lung cancer sequencing studies or large-scale pan-cancer analyses using $q < 0.05$ as cutoff, or present in the COSMIC cancer gene census (August 2015). Next, non-silent variants in these genes were evaluated, and putative driver mutations were identified if they met one of the following requirements: (1) either the exact mutation, the same mutation site or at least three mutations located within 15 bp of the variant were found in COSMIC and (2) if the candidate gene was marked as recessive in COSMIC and the variant was predicted to be deleterious, including stop-gain, frameshift and splicing mutation, and had a SIFT score < 0.05 or a PolyPhen score > 0.995. Then, the putative driver mutations were subjected to Sanger sequencing for validation when adequate DNA was available. The mutated protein structures were generated by Phyre2 web portal. All figures were prepared by using PyMOL (http://pymol. sourceforge.net/).

**Mutational signature analysis.** Both silent and non-silent somatic SNVs were measured to define mutational signatures in the tumors. For each tumor, we extracted the 5′ and 3′ sequence context of each mutation from the hg19 reference genome and the SNVs were categorized into C > A, C > G, C > T, T > A, T > C, and T > G bins according to the type of substitution and then subcategorized into 96 sub-bins according to the nucleotides preceding (5′) and succeeding (3′) the mutated base. To enable comparison with the known signatures based on the Wellcome Trust Sanger Institute Mutational Signature Framework, the R package 'deconstructSigs' was used to statistically quantify the contribution of each signature for each tumor[32].

**Clonality analysis.** PyClone and SciClone were employed to detect subclones. PyClone applied an MCMC method to cluster variants. SciClone focused primarily on variants in copy number neutral, loss of heterozygosity (LOH)-free portions of the genome, which allows for the highest confidence quantification of variant allele frequencies (VAF) and inference of clonality.

**Cancer cell fraction analysis.** For each variant, the expected VAF, given the cancer cell fraction (CCF), was defined as $f(CCF) = \frac{p \times CCF}{CPN_{mut} \times p + CPN_{norm}(1-p)}$, where $CPN_{mut}$ corresponds to the local copy number of the tumor, $CPN_{norm}$ is the local copy number of the matched normal sample and $p$ is the tumor purity. Tumor cellularity was determined on the basis of VAF and segmented copy number data using ABSOLUTE[54], to calculate the CCF of each mutation. Clonal status was defined according to the confidence interval of CCF.

**Neoantigen prediction.** HLA typing was performed using Polysolver[55]. Non-silent mutations served as basis to generate a list of peptides ranging 9–11 amino acids in length with the mutated residues represented in each position. Prediction for binding affinity of every mutant peptide and its corresponding wild-type peptide to the patient's germline HLA alleles was performed using the NetMHCpan algorithm (v3.0)[56]. Candidate neoantigens were identified as those with a predicted mutant peptide binding affinity of < 500 nM and less than that of its corresponding wild-type peptide.

**Cell culture and reagents.** Tumor cell lines were obtained from ATCC or JCRB, where cell characterization (polymorphic short tandem repeat profiling) and contamination were performed. Cells were cultured in RPMI1640 (Invitrogen) supplemented with 10% fetal bovine serum (Gibco). Erlotinib, afatinib, docamatinib, trametinib, dabrafenib, and lapatinib were purchased from Selleck Chemicals. All inhibitors were reconstituted in DMSO (Sigma-Aldrich) at a stock concentration of 10 mM. For cell viability assays, cells were seeded at 100,000 cells per well in growth media supplemented with 10% FBS and 2 mM L-glutamine in 6-well plate, allowed to adhere overnight, and treated with a serial dilution of inhibitors for 1 week. Cells were fixed with formalin, and stained with crystal violet.

**CRISPR-Cas9 knockout**. CRISPR-Cas9 knockout technology was employed to knockout EGFR or KRAS in PC9 or H1944 and H2009 cells, respectively. We constructed LentiCRISPR v2 vector with blasticidin resistance, designed sgRNA and constructed lentiviral CRISPR plasmids according to the previously described protocol[57]. EGFR-sgRNA: 5′-CACCGCTGCGCTCTGCCCGGCGAGT-3′ and 5′-AAACACTCGCCGGGCAGAGCGCAGC-3′; KRAS-sgRNA: 5′-CACCGT-CTCGACACAGCAGGTCAAG-3′ and 5′-AAACCTTGACCTGCTGTGTCGA-GAC-3′. An EGFP-sgRNA (5′-CACCGGAAGTTCGAGGGCGACACCC-3′ and 5′-AAACGGGTGTCGCCCTCGAACTTCC-3′) was used as negative control.

**Western blot**. Cells or tissues were lysed in RIPA buffer (Tris pH 7.4 50 mM, NaCl 150 mM, NP-40 1%, SDS 0.1%, EDTA 2 μM) containing proteinase inhibitors (Roche) and phosphatase inhibitors (Roche). The lysates (20 μg protein) were subjected to SDS-PAGE and western blot. Antibodies against the following proteins were used: phospho-EGFR (Y1068), EGFR, phospho-MEK1/2 (S217/221), MEK1/2, phospho-ERK (T202/Y204), ERK, phospho-AKT (T308), phospho-AKT (S473), AKT, phospho-HER2 (Y1221/1222), HER2, MET, HER4, HER3, PDGFRB, H3 and Rab11 (Cell Signaling Technology). Antibodies against ARAF and BRAF were purchased from Santa Cruz. Anti-RAS antibody was purchased from Millipore. Uncropped images of the immunoblots were shown in Supplementary Fig. 16.

**Immunohistochemistry**. Immunohistochemistry was performed using 5 μm-thick, formalin-fixed, paraffin-embedded tissue sections. Slides were baked, deparaffinized in xylene, passed through graded alcohols, and antigen retrieved with 10 mM citrate buffer, pH 6.0 in a steam pressure cooker. Preprocessed tissues were treated with peroxidase block (Dako) to quench endogenous peroxidase activity, blocked using protein block (Dako), and incubated with indicated antibodies. Slides were then washed in 50 mM Tris-Cl, pH 7.4 and incubated with horseradish peroxidase-conjugated secondary antibody. Immunoperoxidase staining was developed using a 3,3′diaminobenzidine (DAB) chromogen. Slides were counterstained with hematoxylin, dehydrated in graded alcohol and xylene, and coverslipped using mounting solution.

**Tumor xenograft and PDX models**. Tumor cells ($1 \times 10^6$) were mixed with Matrigel (BD Biosciences) and subcutaneously implanted in the dorsal flank of BALB/c nude mice. When tumor sizes reached ~ 150 mm³, animals were randomized into two groups of 10 mice each. One group of mice was treated with vehicle control (0.5% methylcellulose and 0.2% Tween-80), and the other group was treated with trametinib (1 mg per kg per day). Tumor volumes (10 animals per group) were measured with digital caliper and calculated as length × width² × 0.52. The animals were housed in a specific pathogen free (SPF) animal facility in accordance with the Guide for Care and Use of Laboratory Animals and the regulations of the Institutional Animal Care and Use Committee in Ren Ji Hospital. Lung PDX model was established using patient tumor tissues acquired during surgery. Prior written informed consent was obtained from the patient. Experiments were conducted on female BALB/c nude mice aged 6–8 weeks old. Freshly collected tumor samples were cut into small pieces and implanted subcutaneously to the flanks of nude mice. Tumors used in this study were over passage-3 with stable tumor growth. Tumor-bearing mice with tumors at 100–250 mm³ range were selected and randomly divided into four groups of 10 mice each. One group of mice was treated with vehicle control (0.5% methylcellulose and 0.2% Tween-80), and the other three groups were treated with dabrafenib (30 mg per kg per day), trametinib (1 mg per kg per day) and a combination of both drugs, respectively. Tumor volumes (10 animals per group) were measured with digital caliper and calculated as length × width² × 0.52.

**Statistical analysis**. Statistical analysis was performed with GraphPad Prism software. In all experiments, comparisons between two groups were based on two-sided Student's t-test and one-way analysis of variance (ANOVA) was used to test for differences among more groups. P-values of < 0.05 were considered statistically significant.

**Data availability**. The sequencing data have been deposited in NCBI SRA database (http://www.ncbi.nlm.nih.gov/SRA/) under the accession number SRP095985. All other data are included within the article or available from the authors upon request.

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

## Acknowledgements

This work was supported by the National Natural Science Foundation of China (81472537 and 81672714 to G.Z.; 81502597 to Y.J.; 81630073 and 81372189 to W.-Q.G.; 81473232 to Y.S.; 81573018 to L.Z.), the Grants from the State Key Laboratory of Oncogenes and Related Genes (No. 91–15–12 to G.Z.; SB17-06 to M.-C.C.), the Project Funded by China Postdoctoral Science Foundation (2015M580338 to P.M.), the Grant from Shanghai Municipal Commission of Health and Family Planning (20174Y0043 to M.-C.C.) Shanghai Municipal Education Commission-Gaofeng Clinical Medicine Grant Support (20161313 to G.Z.), Collaborative Innovation Center for Translational Medicine at Shanghai Jiao Tong University School of Medicine, the Shanghai Institutions of Higher Learning (Eastern Scholar to G.Z.), Shanghai Rising-Star Program (16QA1403600 to G.Z.), the Incubating Program for Clinical Research and Innovation of Renji Hospital (PYXJS16-001 to Y.J.), Science and Technology Commission of Shanghai Municipality (16JC1405700 to W.-Q.G.), the KC Wong foundation to W.-Q.G., grants from NIH/NCI (R01CA169338, R01CA211052 and R01CA204302 to T.G.B.) and Pew and Stewart Foundations to T.G.B.

## Author contributions

P.M., Y.F. and M.-C.C. designed and conceived the experiments. Y.J. and S.Z. performed the experiments. M.C., J.W., Y.S., L.Z., H.-Z.C., W.-Q.G. and M.W. analyzed the data. Y.Y. and Z.G. established the PDX model and performed the efficacy study. P.M., T.G.B. and G.Z. drafted the manuscript and all authors contributed to writing the manuscript. X.Z. and G.Z. jointly supervised the study.

## Additional information

**Competing interests:** The authors declare no competing financial interests.

