## [Peer Review File · Nature Communications]

Reviewers' Comments:

Reviewer #1:

Remarks to the Author:

NCOMMS-17-04070-T

"Simultaneous evolutionary expansion and constraint of genomic heterogeneity in multifocal lung cancer"

The authors perform detailed whole exome (and some whole genome) DNA sequence analysis of lung adenocarcinomas arising in multiple synchronous lung cancers (MSLCs) in 4 female never smokers. They then compare the mutations representing coding changes in tumors from the same individual. They show by analyses of silent mutations that the lung cancers are indeed independent of one another and not metastatic lesions from the same tumor. Their key conclusions are that despite having different driver mutations the tumors from each patient appear to "converge" on the signaling pathways affected. They provide functional tests in preclinical models for several of the driver mutations found to show oncogenic activity including the ability to rescue CRISPR as9 mediated deletion of several driver mutations in lung cancer cell lines. Several xenograft studies treated with various targeted therapies targeted at the signaling pathways involved were found to respond. Finally, they present data on genomic analyses of lung and breast cancers found in the same patient. While these had entirely different driver mutations they also showed similar signaling pathways affected.

Comments to the Authors:

This is a technically well done genomic analysis of an "experiment of nature" that shows the "expansion" (of genetic changes) and "convergence" (of signaling changes) in MSLCs. This paper is reviewed in the context of a very similar publication in Nature Communication in 2016 (their reference 28) which studied 17 lung adenocarcinomas in 6 patients (4 never smokers and 2 former smokers). The ethnicity of patients studied in both the current paper and the reference 28 paper are likely to both be East Asian. Thus, the genomic findings in the two studies are on almost identical tumors and tumor settings and the main results appear to be very similar. The current study provides some functional data which the other study does not. There are several things that would have greatly improved the current report.

1. The main paper needs a simple table (that currently is given in the supplemental section) on the age, gender, ethnicity, smoking status and presence of well known driver mutations in the tumors from the 4 patients. If other data needs to be moved to the supplemental section to accommodate this that is fine. The reader should not have to go to the supplemental sections to find these key pieces of information.

2. If they had information on MSLCs in at least one smoker their (which is the most common type of lung cancer) their case would have been strengthened.

3. Given their arguments of convergence of pathways, I expected to see multiple immunohistochemical (IHC) stains testing a variety of signaling pathways known to be altered in lung adenocarcinomas (other than ERK) to functionally prove in these patients that signaling pathways actually expressed were indeed constrained. Similarly, IHC stains of adenocarcinoma lineage oncogenes and differentiation markers (like TTF1 and napsin) to see how these compared between the MSLCs would have been very helpful.

4. We know the tumor microenvironment plays a key role in lung cancer pathogenesis including potentially immune responses. So one question is whether the neoantigen load was similar in the different MSLCs and was there any evidence of immune selection similarities or differences in tumors from the same patient?

5. A part of the same analyses, IHC stains for key components of the tumor microenvironment would have been very helpful. Whatever the answers are (similar or different for the MSLCs) they would be important to know. In fact, such information would have huge impact on planning for precision immune treatment regimens as much or perhaps more than for targeted therapy.

6. Do "branched" or multi-region analyses within a tumor show the same kind of convergence? In

the paper's current format, I am not sure what the authors' conclusions are from the analyses presented.

7. While other molecular studies are perhaps the subject of future research, I would have been interested to know the status of epigenetic changes and RNA expression profiles in the different tumors from the same patient. Such information could have also provide answers to the questions posed above such as information on signaling pathways and the tumor micro-environment. The epigenetic studies would give us important information on whether they show the same expansion and convergence that the mutational studies show.

8. Why was the 5th patient reported separately? This seems unclear.

Reviewer #2:

Remarks to the Author:

The authors present a multi-region sequencing analysis of a unique cohort of multifocal NSCLCs. Both the sample set and the functional validation are very interesting. However there are a few major issues that would need to be addressed:

1) The phenotypic/functional convergence between lesions from the same patient is not surprising. Mutations in EGFR (alone) account for 10-35% of NSCLC and the whole pathway could amount to up to 50%. Similarly, KRAS+BRAF+etc can be as high as 35%. Without a large patient cohort, it is not surprising that two or three co-occurring tumours are driven by the same pathway. This issue would be compounded if the patients in the study were selected from a larger study cohort. Hence the conclusions in the manuscript need to be tempered.

2) The 'schematic model' in figure 5e makes very little sense and does not add anything to the discussion.

3) Fig. 2c: Despite the importance of protein expression changes when KRAS, BRAF, MEK1 were modified in HEK293T and LXF-289 cell lines, the same has not been shown in BEAS-2B cell line. This is surprising since BEAS-2B cell line was chosen for two important assays (Fig. 2d and 2e) and to make a strong argument for their case. It will be important to demonstrate the same panel of protein expression changes in modified BEAS-2B cells before moving to in vivo.

4) Fig. 2c: pERK expression seems to be differently regulated in HEK293T and LXF-289 cell lines when D594G modification was introduced. It is surprising to see that the authors performed an experiment using human kidney cell line and made a comparison with lung adenocarcinoma cell line. The latter shows increased kinase activity in lung adenocarcinoma cell line with BRAF D594G modification as assessed by western blot analysis. This difference needs to be discussed in more detail and the authors need to take the increase of pERK expression in BRAF D594G modified LXF-289 cells into consideration.

5) Fig. 2d: It would make a stronger argument if they have performed the colony formation assay in at least KRAS modified BEAS-2B cell line.

6) Fig. 2e: This experiment is lacking a very important point where the comparison between WT KRAS, BRAF, MEK1 and modified versions has not been performed while this has been clearly demonstrated in Fig.2c and 2d. This is critical as the main message of this figure should be the difference between WT KRAS, BRAF, MEK1 and modified versions rather than with empty vector.

**Reviewer #1 Expert in lung cancer genomics:
Comments to the Author**

The authors perform detailed whole exome (and some whole genome) DNA sequence analysis of lung adenocarcinomas arising in multiple synchronous lung cancers (MSLCs) in 4 female never smokers. They then compare the mutations representing coding changes in tumors from the same individual. They show by analyses of silent mutations that the lung cancers are indeed independent of one another and not metastatic lesions from the same tumor. Their key conclusions are that despite having different driver mutations the tumors from each patient appear to “converge” on the signaling pathways affected. They provide functional tests in preclinical models for several of the driver mutations found to show oncogenic activity including the ability to rescue CRISPR as9 mediated deletion of several driver mutations in lung cancer cell lines. Several xenograft studies treated with various targeted therapies targeted at the signally pathways involved were found to respond. Finally, they present data on genomic analyses of lung and breast cancers found in the same patient. While these had entirely different drive mutations they also showed similar signaling pathways affected.

Comments to the Authors: This is a technically well done genomic analysis of an “experiment of nature” that shows the “expansion” (of genetic changes) and “convergence” (of signaling changes) in MSLCs. This paper is reviewed in the context of a very similar publication in Nature Communication in 2016 (their reference 28) which studied 17 lung adenocarcinomas in 6 patients (4 never smokers and 2 former smokers). The ethnicity of patients studied in both the current paper and the reference 28 paper are likely to both by East Asian. Thus, the genomic findings in the two studies are on almost identical tumors and tumor settings and the main results appear to be very similar. The current study provides some functional data which the other study does not. There are several things that would have greatly improved the current report.

1. The main paper needs a simple table (that currently is given in the supplemental section) on the age, gender, ethnicity, smoking status and presence of well known driver mutations in the tumors from the 4 patients. If other data needs to be moved to the supplemental section to accommodate this that is fine. The reader should not have to go to the supplemental sections to find these key pieces of information.

We thank the reviewer’s suggestions, and have added a table (originally Supplementary Table 1) to provide the clinical information of the 4 patients with MSLC. The driver mutations for each tumor are shown in Fig. 3a.

2. If they had information on MSLCs in at least one smoker their (which is the most common type of lung cancer) their case would have been strengthened.

We agree with the reviewer that it would be nice to have included more MSLCs in our cohort, especially those with smoking history. Unfortunately, however, we found that most MSLC patients eligible for surgery were non-smokers with early-stage disease. In addition, it was a priority to preserve intact tumor tissues for accurate pathological diagnosis, which made sample collection of MSLCs particularly difficult. Consequently, we were not yet able to obtain appropriate MSLC specimens from smokers. An expanded, multicenter clinical investigation of MSLCs is underway and we hope to address this question in the future.

3. Given their arguments of convergence of pathways, I expected to see multiple immunohistochemical (IHC) stains testing a variety of signaling pathways known to be altered in lung adenocarcinomas (other than ERK) to functionally prove in these patients that signaling pathways actually expressed were indeed constrained. Similarly, IHC stains of adenocarcinoma lineage oncogenes and differentiation markers (like TTF1 and napsin) to see how these compared between the MSLCs would have been very helpful.

Following the reviewer's advice, we have performed immunohistochemical staining of phospho-AKT (T308), Napsin A and TTF-1. These data have been included in the revised manuscript (Supplementary Fig. 13a).

4. We know the tumor microenvironment plays a key role in lung cancer pathogenesis including potentially immune responses. So one question is whether the neoantigen load was similar in the different MSLCs and was there any evidence of immune selection similarities or differences in tumors from the same patient?

We thank the reviewer for the great suggestions. Neoantigens were predicted for each tumor of MSLCs using NetMHCpan algorithm, which revealed distinct repertoires and unequal load of neoantigens associated with different lesions (Supplementary Fig. 9b; Supplementary Table 5). These results implied functionally heterogeneous tumor-associated lymphocytes and potentially differential clinical responses to immune treatment regimens among MSLC loci. We have included these data in the revised manuscript.

5. A part of the same analyses, IHC stains for key components of the tumor microenvironment would have been very helpful. Whatever the answers are (similar or different for the MSLCs) they would be important to know. In fact, such information would have huge impact on planning for precision immune treatment regimens as much or perhaps more than for targeted therapy.

To address the heterogeneity of tumor microenvironment, CD3 IHC was conducted to probe the T cell population. Interestingly, consistent with a recent report, all tumor sections of MSLCs exhibited positive staining of CD3 (Supplementary Fig. 9a),

suggesting that T cells might already infiltrate into tumors even in early-stage lung cancer. The functional diversity of these immune cells warrants further investigations in future studies.

6. Do “branched” or multi-region analyses within a tumor show the same kind of convergence? In the paper’s current format, I am not sure what the authors’ conclusions are from the analyses presented.

As shown in Fig. 2f and Fig. 2g, the putative driver alterations for each region were mostly present in all sequenced loci, and thus represent truncal/early somatic mutations. Since there were very few driver events on the branches, we did not observe the same kind of convergent evolution as in MSLCs. Previous reports have indicated that most known driver mutations occur early in lung tumor evolution. On the other hand, studies in clear cell renal cell carcinoma (Nat Genet. 2014 Mar;46(3):225-33.) and glioma (Nat Genet. 2015 May;47(5):458-68.) did indeed find typical examples of parallel mutations with multi-region analyses. It should be noted that the subclones in those studies originated from the same tumor progenitors, and therefore were fundamentally different from multiple synchronous thoracic lesions which were completely independent as described in our manuscript.

7. While other molecular studies are perhaps the subject of future research, I would have been interested to know the status of epigenetic changes and RNA expression profiles in the different tumors from the same patient. Such information could have also provide answers to the questions posed above such as information on signaling pathways and the tumor micro-environment. The epigenetic studies would give us important information on whether they show the same expansion and convergence that the mutational studies show.

We agree with reviewer that the status of epigenetic changes and RNA expression profiles in different tumors from the same patient would be quite informative to illuminate the evolutionary principles of MSLC. However, with very limited tumor materials available, we were only able to perform genomic DNA sequencing as the first step to delineate the molecular underpinnings of MSLC development. Epigenetic or transcriptomic analyses require high-quality clinical samples and yet-to-be-improved quantitative approach to deal with small cohort size, which, at the moment, is extremely challenging and beyond the scope of our current project. We appreciate the reviewer for the insightful comments, but feel that such studies could be subjects of future reports.

8. Why was the 5th patient reported separately? This seems unclear.

We used the first 4 MSLC patients (RJLC1-4) as the discovery cohort, and the last 3 (RJLC5-7) patients as the validation cohort. We collected fresh tumor samples upon surgery from each lesion of RJLC1-4, and these tissues were subjected to whole-exome sequencing for the initial genomic characterization of MSLCs. RJLC5-6 harbored a tumor in breast and a tumor in lung, and thus represented a different, but related, scenario. RJLC7 was a special case, who underwent surgical resection of two MSLCs in her right

lobes and received gefitinib treatment to control the remaining tumor nodes in the left upper lobe. Therefore, these analyses provided several complementary lines of evidence to support the evolutionary principle of MSLC development that we proposed.

**Reviewer #2 Expert in tumour heterogeneity:
Comments to the Author**

The authors present a multi-region sequencing analysis of a unique cohort of multifocal NSCLCs. Both the sample set and the functional validation are very interesting. However there are a few major issues that would need to be addressed:

1) The phenotypic/functional convergence between lesions from the same patient is not surprising. Mutations in EGFR (alone) account for 10-35% of NSCLC and the whole pathway could amount to up to 50%. Similarly, KRAS+BRAF+etc can be as high as 35%. Without a large patient cohort, it is not surprising that two or three co-occurring tumours are driven by the same pathway. This issue would be compounded if the patients in the study were selected from a larger study cohort. Hence the conclusions in the manuscript need to be tempered.

The reviewer raised a critical point. It is notable that except EGFR and KRAS, other driver mutations identified in our study were diverse and relatively rare in NSCLC. These included BRAF^{D594G}, c-MET^{ΔE14}, ARAF^{S214C} or MEK1^{E102-I103 del} and HER2^{YMVA}, which led us to believe that evolutionary constraints caused this striking convergence phenomenon. However, as the reviewer suggested, we cannot completely rule out the possibility that the functional recurrent patterns in some cases may be due to coincidence, and we have toned down our conclusions in the revised manuscript.

2) The 'schematic model' in figure 5e makes very little sense and does not add anything to the discussion.

We apologize for the confusion caused by insufficient explanation. To summarize the findings of our detailed genomic analyses in MSLCs, we proposed this schematic model in Fig. 4e for open discussion, in which we believed that simultaneous evolutionary expansion and constraint of genomic heterogeneity collaboratively shaped tumorigenic process and clonal architecture. While many studies have focused on the substantial tumor heterogeneity as a result of stochastic mutations, very little attention has been given to the evolutionary constraints during tumorigenesis. Therefore, it might be important, in our opinion, to highlight the dueling evolutionary forces that control the formation of fittest tumor clones, in order to thoroughly understand tumor evolution and rationally optimize therapeutic strategies.

3) Fig. 2c: Despite the importance of protein expression changes when KRAS, BRAF, MEK1 were modified in HEK293T and LXF-289 cell lines, the same has not been shown

in BEAS-2B cell line. This is surprising since BEAS-2B cell line was chosen for two important assays (Fig. 2d and 2e) and to make a strong argument for their case. It will be important to demonstrate the same panel of protein expression changes in modified BEAS-2B cells before moving to in vivo.

We thank the reviewer for the great point. The impact of wild-type or mutant KRAS, BRAF, MEK1 expression on downstream MAPK pathway in BEAS-2B cells has been determined and the results were shown in revised Fig. 2c. All these putative driver oncogenic mutations led to ERK hyperactivation.

4) Fig. 2c: pERK expression seems to be differently regulated in HEK293T and LXF-289 cell lines when D594G modification was introduced. It is surprising to see that the authors performed an experiment using human kidney cell line and made a comparison with lung adenocarcinoma cell line. The latter shows increased kinase activity in lung adenocarcinoma cell line with BRAF D594G modification as assessed by western blot analysis. This difference needs to be discussed in more detail and the authors need to take the increase of pERK expression in BRAF D594G modified LXF-289 cells into consideration.

D594 resides in the highly conserved DFG motif of BRAF kinase domain. As a result, mutations of this residue, including D594G or D594N described in our manuscript, are catalytically inactive, which is consistent with reduced phospho-ERK in HEK293T cells associated with BRAF^{D594G} as compared to BRAF^{WT}. Counterintuitively, D594 mutants have been frequently discovered in various human cancers (<http://www.cbioportal.org/>). Heidorn et al proposed that these kinase-dead mutants was proficient in binding with CRAF and consequently driving MEK/ERK activation in the presence of oncogenic RAS (Cell. 2010 Jan 22;140(2):209-21.). We speculate that this scaffold mechanism of BRAF^{D594G} to promote RAS-CRAF signaling underlies the increased phospho-ERK and transforming phenotype in BEAS-2B or LXF-289 cells. Therefore, the effects of BRAF^{D594G} on MAPK pathway activity are context-dependent, and we have discussed this point in more detail in the revised manuscript.

5) Fig. 2d: It would make a stronger argument if they have performed the colony formation assay in at least KRAS modified BEAS-2B cell line.

We have attempted to perform the colony formation assay in BEAS-2B and LXF-289 cells overexpressing KRAS^{WT} or KRAS^{G12D}. Consistent with previous reports, we found that KRAS^{G12D} expression triggered cellular senescence in vitro, which confounded the evaluation of its oncogenic potential using this assay (Please see the images shown below). Considering that KRAS^{G12D} is a well-characterized oncogene, and that our focus was two novel candidate driver alterations (BRAF^{D594G} and MEK1^{E102-I103 del}), we did not discuss these results in our manuscript.

6) Fig. 2e: This experiment is lacking a very important point where the comparison between WT KRAS, BRAF, MEK1 and modified versions has not been performed while this has been clearly demonstrated in Fig.2c and 2d. This is critical as the main message of this figure should be the difference between WT KRAS, BRAF, MEK1 and modified versions rather than with empty vector.

We have performed the xenograftation experiment for BEAS-2B cells expressing wild-type KRAS, BRAF, MEK1, and found that none of these cells were able to form palpable tumors within the same time frame (Please see the images shown below). The manuscript has been revised to include these controls (Fig. 2e).

Reviewers' Comments:

Reviewer #1:

Remarks to the Author:

The authors have responded appropriately to all of the reviewers' comments including providing additional data.

Reviewer #2:

Remarks to the Author:

The authors addressed all my concerns.